# LeOCLR: Leveraging Original Images for Contrastive Learning of Visual Representations

**Mohammad Alkhalefi**  *m.alkhalefi1.21@abdn.ac.uk*
*Department of Computing Science*
*University of Aberdeen*

**Georgios Leontidis**  *georgios.leontidis@abdn.ac.uk*
*Department of Computing Science & Interdisciplinary Institute*
*University of Aberdeen*

**Mingjun Zhong**  *mingjun.zhong@abdn.ac.uk*
*Department of Computing Science*
*University of Aberdeen*

**Reviewed on OpenReview:** *https://openreview.net/forum?id=y8qGOvUn1r*

## Abstract

Contrastive instance discrimination methods outperform supervised learning in downstream tasks such as image classification and object detection. However, these methods rely heavily on data augmentation during representation learning, which can lead to suboptimal results if not implemented carefully. A common augmentation technique in contrastive learning is random cropping followed by resizing. This can degrade the quality of representation learning when the two random crops contain distinct semantic content. To tackle this issue, we introduce LeOCLR (Leveraging Original Images for Contrastive Learning of Visual Representations), a framework that employs a novel instance discrimination approach and an adapted loss function. This method prevents the loss of important semantic features caused by mapping different object parts during representation learning. Our experiments demonstrate that LeOCLR consistently improves representation learning across various datasets, outperforming baseline models. For instance, LeOCLR surpasses MoCo-v2 by 5.1% on ImageNet-1K in linear evaluation and outperforms several other methods on transfer learning and object detection tasks.

## 1 Introduction

Self-supervised learning (SSL) approaches based on instance discrimination (Chen et al., 2020b; Chen & He, 2021; Chen et al., 2020a; Misra & Maaten, 2020; Grill et al., 2020) heavily rely on data augmentations, such as random cropping, rotation, and colour Jitter, to build invariant representation for all the instances in the dataset. To do so, the two augmented views (positive pairs) for the same instance are attracted in the latent space while avoiding collapse to the trivial solution (representation collapse). These approaches have proven efficient in learning useful representations by using different downstream tasks, e.g., image classification and object detection, as proxy evaluations for representation learning. However, these strategies ignore the important fact that the augmented views may have different semantic content due to random cropping, which can lead to a degradation in visual representation learning (Song et al., 2023; Zhang et al., 2022; Liu et al., 2020; Mishra et al., 2021). On the one hand, creating positive pairs through random cropping and encouraging the model to make them similar based on the shared information in the two views makes the SSL task harder, ultimately improving representation quality (Mishra et al., 2021; Chen et al., 2020a). In addition, random cropping followed by resizing leads model representation to capture information for the

object from varying aspect ratios and induce occlusion invariance (Purushwalkam & Gupta, 2020). On the other hand, minimizing the feature distance in the latent space (i.e., maximizing similarity) between views containing distinct semantic concepts tends to result in the loss of valuable image information (Purushwalkam & Gupta, 2020; Zhang et al., 2022; Song et al., 2023).

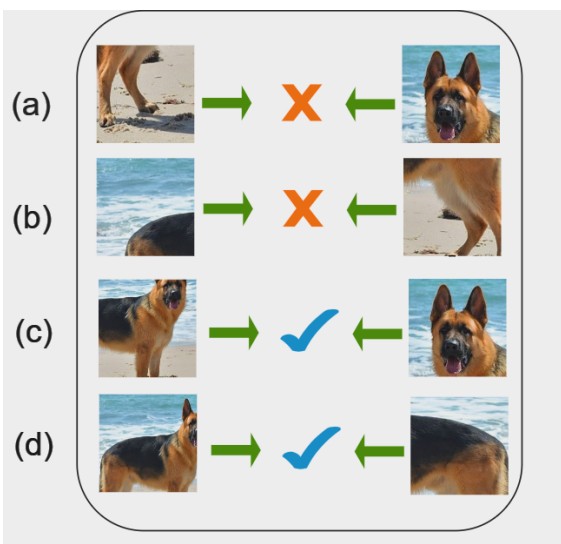

Figure 1: Examples of positive pairs that might be created by random cropping and resizing.

Figure 1 (a and b) show examples of incorrect semantic positive pairs (i.e., positive pairs that contain mismatched semantic information for the same object) that may result from random cropping. In example (a), when the model is forced to align the representations of a dog's head and leg closer in the latent space, it may discard important semantic features. This happens because the model bases its representations on the shared region between the two views. If this shared region does not contain semantically consistent information, the representations become trivial. For random cropping to be effective and achieve occlusion invariance, the shared region must convey the same semantic information in both views. In contrast, Figure 1 (c and d) shows positive pairs where the shared region contains similar semantic content. For example, in (c), both views contain the dog's head, which helps the model capture features of the dog's head across different scales and angles. It is worth noting, though, that there is value in contrasting pairs that might include different semantic information about the same object (e.g., a dog in our case), as it can aid in learning global features.

As the examples show, creating random crops for one-centric object does not guarantee obtaining correct semantic pairs. This consideration is important for improving representation learning. Instance discrimination SSL approaches, such as MoCo-v2 (Chen et al., 2020b) and SimCLR (Chen et al., 2020a), encourage the model to bring positive pairs, i.e., two views for the same instance, closer in the latent space regardless of their semantic content (Xiao et al., 2021; Zhang et al., 2022). This could limit the model's ability to learn representations of different object parts and may impair its capability to learn representations of semantic features (Song et al., 2023; Zhang et al., 2022) (see Figure 2 (left)).

It has been shown that undesirable views containing different semantic content may be unavoidable when employing random cropping (Song et al., 2023). Therefore, we need a method to train the model on different object parts to develop robust representations against natural transformations, such as scale and occlusion, rather than simply pulling the augmented views together indiscriminately (Mishra et al., 2021). This issue should be addressed, as the performance of downstream tasks depends on high-quality visual representations learned through self-supervised learning (Alkhalefi et al., 2024; Donahue et al., 2014; Manová et al., 2023; Girshick et al., 2014; Zeiler & Fergus, 2014; Kim & Walter, 2017; Zhang & Ma, 2022; Xiao et al., 2020).

Our work introduces a new instance discrimination SSL approach to avoid forcing the model to make similar representations for the two positive views regardless of their semantic content. As shown in Figure 2 (right), we include the original image $X$ in the training process, as it encompasses all the semantic features of the views $X^1$ and $X^2$. In our approach, the positive pairs (i.e., $X^1$ and $X^2$) are pulled to the original image $X$ in the latent space, unlike contrastive SOTA approaches like SimCLR (Chen et al., 2020a) and MoCo-v2 (Chen et al., 2020b), which attract the two views to each other. This training method ensures that the information in the shared region between the attracted views $(X, X^1)$ and $(X, X^2)$ is semantically correct. As a result, the model learns better semantic features by aligning with the correct semantic content, rather than matching random views that may contain different semantic information. In other words, the model learns the representations of diverse parts of the object because the shared region includes correct semantic parts of the object. This is contrary to other approaches, which may discard important semantic features by incorrectly mapping object parts in positive pairs. Our contributions are as follows:

- We introduce a new contrastive instance discrimination SSL method, LeOCLR, designed to reduce the loss of semantic features caused by mapping two random views that are semantically inconsistent.

- We show that our approach improves visual representation learning in contrastive instance discrimination SSL, outperforming state-of-the-art (SOTA) methods across various downstream tasks.

- We demonstrate that our approach consistently enhances visual representation learning for contrastive instance discrimination across different datasets and contrastive mechanisms.

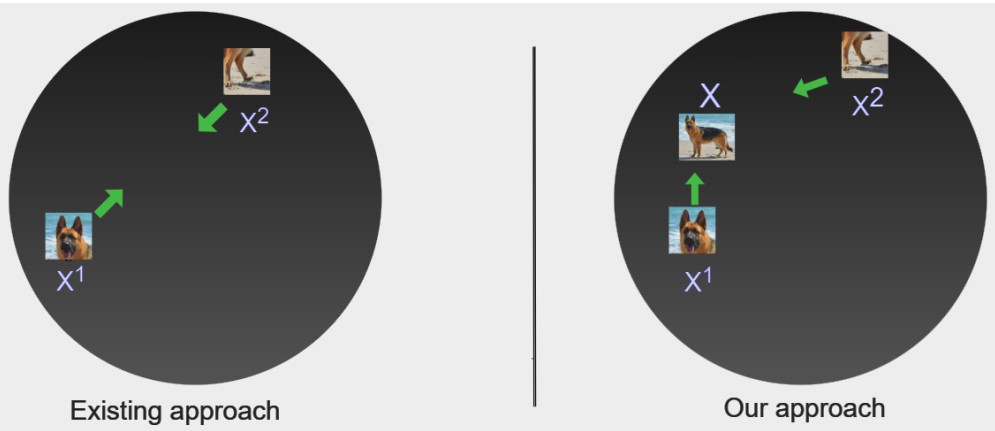

Figure 2: The figure on the left shows the embedding space of established approaches (Chen et al., 2020a;b) where the two views are attracted to each other regardless of their content. In contrast, the figure on the right illustrates our approach, which clusters the two random views together with the original image in the embedding space.

## 2 Related Work

SSL approaches are divided into two broad categories: contrastive and non-contrastive learning. While all these methods aim to bring positive pairs closer in latent space, each uses a different strategy to avoid representation collapse. This section provides a brief overview of some of these approaches, and we encourage readers to refer to the respective papers for more details.

**Contrastive Learning:** Instance discrimination methods, such as SimCLR, MoCo, and PIRL (Chen et al., 2020a; He et al., 2020; Chen et al., 2020b; Misra & Maaten, 2020) employ a similar idea. They attract the positive pairs together and push the negative pairs apart in the embedding space, albeit through a different mechanism. SimCLR (Chen et al., 2020a) uses an end-to-end approach where a large batch size is used for the negative examples and both encoders' parameters in the Siamese network are updated together.

PIRL (Misra & Maaten, 2020) uses a memory bank for negative examples and both encoders' parameters are updated together. MoCo (Chen et al., 2020b; He et al., 2020) takes a momentum contrastive approach where the query encoder is updated during backpropagation, and, in turn, updates the key encoder. Negative examples are stored in a separate dictionary, allowing for the use of large batch sizes.

**Non-Contrastive Learning:** Non-contrastive approaches use only positive pairs to learn visual representations employing different methods to avoid representation collapse. The first category is clustering-based methods, where samples with similar features are assigned to the same cluster. DeepCluster (Caron et al., 2018) uses pseudo-labels from the previous iteration, which makes it computationally expensive and hard to scale. SWAV (Caron et al., 2020) addresses this issue by using online clustering, though it requires determining the correct number of prototypes. The second category involves knowledge distillation. Methods like BYOL (Grill et al., 2020) and SimSiam (Chen & He, 2021) draw on knowledge distillation techniques, where a Siamese network consists of an online encoder and a target encoder. The target network parameters are not updated during backpropagation. Instead, only the online network parameters are updated while being encouraged to predict the representation of the target network. Despite their promising results, how these methods avoid collapse is not yet fully understood. Inspired by BYOL, Self-distillation with no labels (DINO) (Caron et al., 2021) uses centering and sharpening, along with a different backbone (ViT), which allows it to outperform other self-supervised methods while being more computationally efficient. Another approach, Bag of visual words (Gidaris et al., 2020; 2021), uses a teacher-student framework inspired by natural language processing (NLP) to avoid representation collapse. The student network predicts a histogram of the features for augmented images, similar to the teacher network's histogram. The last category is information maximization. Methods like Barlow twins (Zbontar et al., 2021) and VICReg (Bardes et al., 2021) do not require negative examples, stop gradient or clustering. Instead, they use regularization to avoid representation collapse. The objective function of these methods tries to eliminate the redundant information in the embeddings by making the correlation of the embedding vectors closer to the identity matrix. While these methods show promising results, they have limitations, such as the representation learning being sensitive to regularization and decreased effectiveness if certain statistical properties are missing in the data.

**Instance Discrimination With Multi-Crops:** Different SSL approaches introduce multi-crop methods to enable models to learn visual representations of the object from various perspectives. However, when generating multiple cropped views from the same object instance, these views may contain distinct semantic information. To address this issue, LoGo (Zhang et al., 2022) creates two random global crops and $N$ local views. They assume that global and local views of an object share similar semantic content, increasing similarity between these views. At the same time, they argue that different local views have distinct semantic content, thus reducing similarity among them. SCFS (Song et al., 2023) proposes a different solution for handling unmatched semantic views, searching for semantic-consistent features between the contrasted views. CLSA (Wang & Qi, 2022) generates multi-crops and applies both strong and weak augmentations to them, using distance divergence loss to enhance instance discrimination in representation learning. Previous approaches assume that global views contain similar semantic content and treat them indiscriminately as positive pairs. However, our approach suggests that global views may contain incorrect semantic pairs due to random cropping, as shown in Figure 1. Therefore, we aim to attract the two global views to the original (intact and uncropped) image, which fully encapsulates the semantic features of the crops.

## 3 Methodology

Mapping incorrect semantic positive pairs (i.e., positive pairs containing different semantic views) leads to the loss of semantic features, which degrades model representation learning (Mishra et al., 2021; Purushwalkam & Gupta, 2020; Song et al., 2023). To address this, we introduce a new contrastive instance discrimination SSL strategy called LeOCLR. Our approach aims to capture meaningful features from two random positive pairs, even if they contain different semantic content, to enhance representation learning. To achieve this, it is crucial to ensure that the information in the shared region between the attracted views is semantically correct. This is because the choice of views controls the information captured by the representations learned in contrastive learning (Tian et al., 2020). Since we cannot guarantee that the shared region between the two views includes correct semantic parts of the object, we propose to involve the original image in the training

process. The original image $X$, which remains uncropped (i.e., no random cropping), encompasses all the semantic features of the two cropped views, $X^1$ and $X^2$.

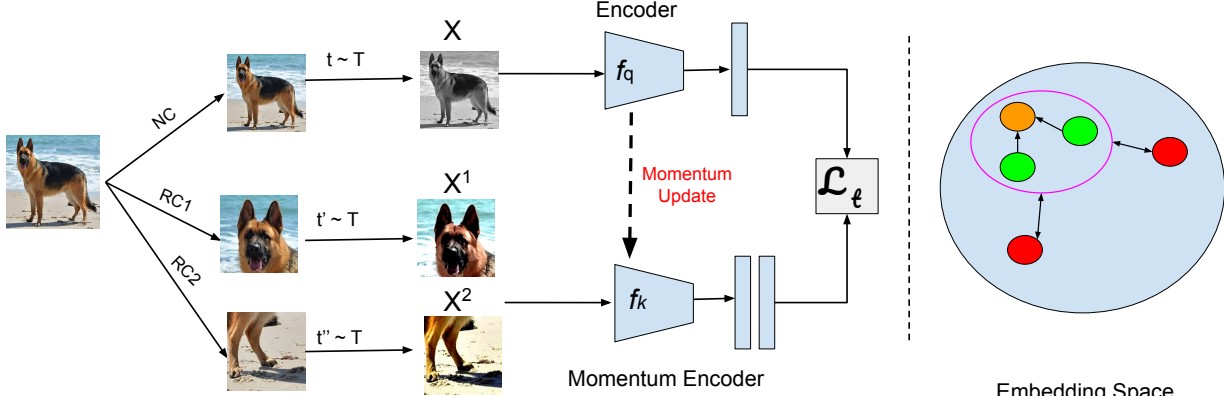

Figure 3: LeOCLR: Overview of the proposed approach. The left part illustrates that the original image $X$ is not cropped (NC), but is resized to $224 \times 224$, before applying transformations. The other views ($X^1$ and $X^2$) are randomly cropped (RC1 and RC2) and resized to $224 \times 224$, followed by the application of transformations. The embedding space of our approach is depicted on the right side of the Figure.

As shown in Figure 3 (left), our methodology generates three views ($X$, $X^1$, and $X^2$). The original image ($X$) is resized without cropping, while the other views ($X^1$ and $X^2$) are randomly cropped and resized. All views are then randomly augmented to prevent the model from learning trivial features. We use data augmentations similar to those employed in MoCo-v2 (Chen et al., 2020b). The original image ($X$) is encoded by the encoder $f_q$, while the two views ($X^1$, $X^2$) are encoded by a momentum encoder $f_k$, whose parameters are updated using the following formula:

$$\theta_k \leftarrow m\theta_k + (1 - m)\theta_q \tag{1}$$

where $m$ is a coefficient set to 0.999, $\theta_q$ are encoder parameters of $f_q$ updated through backpropagation, and $\theta_k$ are momentum encoder parameters of $f_k$ updated by $\theta_q$. Finally, the objective function forces the model to pull both views ($X^1, X^2$) closer to the original image ($X$) in the embedding space, while pushing apart all other instances, as illustrated in Figure 3 (right).

### 3.1 Loss function

Firstly, we briefly describe the loss function of MoCo-v2 (Chen et al., 2020b), since we use momentum contrastive learning in our approach. We will then explain our modification to the loss function.

$$\ell(u, v^+) = -\log \frac{\exp(u \cdot v^+/\tau)}{\sum_{n=0}^{N} \exp(u \cdot v_n/\tau)}, \tag{2}$$

where similarity is measured by the dot product. The objective function increases the similarity between the positive pairs ($u \cdot v^+$) by bringing them closer in the embedding space, while pushing apart all the negative samples ($v_n$) in the dictionary to avoid representation collapse. $\tau$ is the temperature hyperparameter of softmax. In our approach, we increase the similarity between the original image (i.e., query's feature representation) $u = f_q(x)$ with the positive pair (i.e., key's feature representation) $v^+ = f_k(x^i)$ ($i = 1, 2$) and push apart all the negative examples ($v_n$). Therefore the total loss for the mini-batch is:

$$l_t = \sum_{i=1}^{N} \ell(u_i, sg(v_i^1)) + \ell(u_i, sg(v_i^2)) \tag{3}$$

where $sg(.)$ denotes the stop-gradient trick, which is crucial for avoiding representation collapse. As shown in Equation 3, the total loss $l_t$ attracts the two views ($v_i^1$ and $v_i^2$) to their original instance $u_i$. This allows the model to capture semantic features from the two random views, even if they contain distinct semantic information. Our approach captures better semantic features than previous contrastive approaches (Chen et al., 2020a;b; He et al., 2020), as we ensure that the shared region between the attracted views contains correct semantic information. Since the original image contains all parts of the object, any part contained in the random crop is also present in the original image. Therefore, when we pull the original image and the two random views closer in the embedding space, the model learns representations of the different parts, creating an occlusion-invariant representation of the object across varying scales and angles. This is in contrast to previous approaches, which pull the two views together in the embedding space regardless of their semantic content, leading to the loss of semantic features (Liu et al., 2020; Purushwalkam & Gupta, 2020; Song et al., 2023) (see Algorithm 1 for the implementation of our approach).

---

**Algorithm 1** Proposed Approach

---

1: **for** $X$ in *dataloader* **do**
2:     $X^1$, $X^2 =$ rc(X)                                    ▷ random crop first and second views
3:     $X,X^1,X^2 =$ augment$(X,X^1,X^2)$                    ▷ apply random augmentation for all the views
4:     $X = f_q(X)$                                          ▷ encode the original image
5:     $X^1 = f_k(X^1)$                                      ▷ encode the first view by momentum encoder
6:     $X^2 = f_k(X^2)$                                      ▷ encode the second view by momentum encoder
7:     $loss1 = \ell(X,X^1)$                                 ▷ computed as shown in *eq*.1
8:     $loss2 = \ell(X,X^2)$                                 ▷ computed as shown in *eq*.1
9:     $l_t = loss1 + loss2$                                 ▷ computed the total loss as shown in *eq*.2
10: **end for**
11:
12: **def** rc(x):
13:     $x=$ T.RandomResizedCrop(224,224)                    ▷ T is transformation from torchvision module
14:     return $x$

---

Equation 3 and Algorithm 1, illustrate the key differences between our approach and previous multi-crop approaches, such as CLSA (Wang & Qi, 2022), SCFC (Song et al., 2023) , and DINO (Caron et al., 2021). The key differences are as follows:

- Previous approaches assume that two global views contain the same semantic information, encouraging the model to focus on similarities and create similar representations for both views. In contrast, our approach uses the original images instead of global views, as we argue that global views may contain incorrect semantic information for the same object. While they may help capture some global features, this could limit the model's ability to learn more universally useful semantic features, ultimately affecting performance.

- Previous approaches use several local random crops, which might be time- and memory-intensive, while our approach uses only two random crops (Caron et al., 2020; Wang & Qi, 2022).

- Our objective function employs different methods to enhance the model's visual representation learning. We encourage the model to make the two random crops similar to the original image, which contains the semantic information for all random crops while avoiding forcing the two crops to have similar representations if they do not share similar semantic information. This approach differs from previous methods, which encourage all crops (global and local) to have similar representations regardless of their semantic information. As a result, although useful for learning some global features, those methods may discard relevant semantic information, potentially hindering the transferability of the resulting representations to downstream tasks.

## 4   Experiments and Results

**Datasets:** We conducted multiple experiments on three datasets: STL-10 "unlabeled" with 100K training images (Coates & Ng, 2011), CIFAR-10 with 50K training images (Krizhevsky, 2009), and ImageNet-1K with 1.28M training images (Russakovsky et al., 2015).

**Training Setup:** We used ResNet50 as the backbone, and the model was trained with the SGD optimizer, with a weight decay 0.0001, momentum of 0.9, and an initial learning rate of 0.03. The mini-batch size was 256, and the model was trained for up to 800 epochs on ImageNet-1K.

**Evaluation:** We used different downstream tasks to evaluate the LeOCLR's representation learning against leading SOTA approaches on ImageNet-1K: linear evaluation, semi-supervised learning, transfer learning, and object detection. For linear evaluation, we followed the standard evaluation protocol (Chen et al., 2020a; He et al., 2020; Huynh et al., 2022; Dwibedi et al., 2021), where a linear classifier was trained for 100 epochs on top of a frozen backbone pre-trained with LeOCLR. The ImageNet-1K training set was used to train the linear classifier from scratch, with random cropping and left-to-right flipping augmentations. Results are reported on the ImageNet-1K validation set using a center crop ($224 \times 224$). In the semi-supervised setting, we fine-tuned the network for 60 epochs using 1% labeled data and 30 epochs using 10% labeled data. Also, we evaluated the learned features on smaller datasets, such as CIFAR (Krizhevsky, 2009), and fine-grained datasets (Krause et al., 2013; Parkhi et al., 2012; Berg et al., 2014), using transfer learning. Finally, we use the PASCAL VOC (Everingham et al., 2010) dataset for object detection.

**Comparing with SOTA Approaches:** We used vanilla MoCo-v2 (Chen et al., 2020b) as a baseline to compare with our approach across different benchmark datasets, given our use of a momentum contrastive learning framework. Additionally, we compared our approach with other SOTA methods on the ImageNet-1K dataset.

Table 1: Comparisons between our approach LeOCLR and SOTA approaches on ImageNet-1K.

| Approach | Epochs | Batch | Accuracy |
|---|---|---|---|
| MoCo-v2 (Chen et al., 2020b) | 800 | 256 | 71.1% |
| BYOL (Grill et al., 2020) | 1000 | 4096 | 74.4% |
| SWAV (Caron et al., 2020) | 800 | 4096 | 75.3% |
| SimCLR (Chen et al., 2020a) | 1000 | 4096 | 69.3% |
| HEXA (Li et al., 2020) | 800 | 256 | 71.7% |
| SimSiam (Chen & He, 2021) | 800 | 512 | 71.3% |
| VICReg (Bardes et al., 2021) | 1000 | 2048 | 73.2% |
| MixSiam (Guo et al., 2021) | 800 | 128 | 72.3% |
| OBoW (Gidaris et al., 2021) | 200 | 256 | 73.8% |
| DINO (Caron et al., 2021) | 800 | 1024 | 75.3% |
| Barlow Twins (Zbontar et al., 2021) | 1000 | 2048 | 73.2% |
| CLSA (Wang & Qi, 2022) | 800 | 256 | 76.2% |
| RegionCL-M (Xu et al., 2022) | 800 | 256 | 73.9% |
| UnMix (Shen et al., 2022) | 800 | 256 | 71.8% |
| HCSC (Guo et al., 2022) | 200 | 256 | 73.3% |
| UniVIP (Li et al., 2022) | 300 | 4096 | 74.2% |
| HAIEV (Zhang & Ma, 2022) | 200 | 256 | 70.1% |
| SCFS (Song et al., 2023) | 800 | 1024 | 75.7% |
| LeOCLR (*ours*) | 800 | 256 | **76.2%** |

Table 1 presents the linear evaluation of our approach compared to other SOTA methods. As shown, our approach outperforms all others, surpassing the baseline (i.e., vanilla MoCo-v2) by 5.1%. This supports our hypothesis that while two global views can capture some global features, they may also contain different semantic information for the same object (e.g., a dog's head versus its leg), which should be considered to improve representation learning. The observed performance gap (i.e., the difference between vanilla MoCo-v2 and LeOCLR) demonstrates that mapping pairs with divergent semantic content hinders representation learning and affects the model's performance in downstream tasks.

**Semi-Supervised Learning on ImageNet-1K:** In this part, we evaluate the performance of LeOCLR under a semi-supervised setting. Specifically, we use 1% and 10% of the labeled training data from ImageNet-1K for fine-tuning, following the semi-supervised protocol introduced in SimCLR (Chen et al., 2020a). The top-1 accuracy, reported in Table 2 after fine-tuning with 1% and 10% of the training data, demonstrates LeOCLR's superiority over all compared methods. This can be attributed to LeOCLR's improved representation learning capabilities, especially compared to other SOTA methods.

Table 2: Semi-supervised training results on ImageNet-1K: Top-1 performances are reported for fine-tuning a pre-trained ResNet-50 with the ImageNet-1K 1% and 10% datasets.
*\* denotes the results are reproduced in this study.*

| Approach\Fraction | ImageNet-1K 1% | ImageNet-1K 10% |
|---|---|---|
| MoCo-v2 (Chen et al., 2020b) * | 47.6% | 64.8% |
| SimCLR (Chen et al., 2020a) | 48.3% | 65.6% |
| BYOL(Grill et al., 2020) | 53.2% | 68.8% |
| SWAV (Caron et al., 2020) | 53.9% | 70.2% |
| DINO (Caron et al., 2021) | 50.2% | 69.3% |
| RegionCL-M (Xu et al., 2022) | 46.1% | 60.4% |
| SCFS (Song et al., 2023) | 54.3% | 70.5% |
| LeOCLR (*ours*) | **62.8%** | **71.5%** |

**Transfer Learning on Downstream Tasks:** We evaluate our self-supervised pretrained model using transfer learning by fine-tuning it on small datasets such as CIFAR (Krizhevsky, 2009), Stanford Cars (Krause et al., 2013), Oxford-IIIT Pets (Parkhi et al., 2012), and Birdsnap (Berg et al., 2014). We follow the transfer learning procedures outlined in (Chen et al., 2020a; Grill et al., 2020) to find optimal hyperparameters for each downstream task. As shown in Table 3, our approach, LeOCLR, outperforms all compared approaches on various downstream tasks. This demonstrates that our model learns useful semantic features, enabling it to generalize better to unseen data in different downstream tasks compared to other approaches. Our method preserves the semantic features of the given objects, thereby improving the model's representation learning capabilities. As a result, it is more effective at extracting important features and predicting correct classes on transferred tasks.

Table 3: Transfer learning results from ImageNet-1K with the standard ResNet-50 architecture.
*\* denotes the results are reproduced in this study.*

| Approach | CIFAR-10 | CIFAR-100 | Car | Birdsnap | Pets |
|---|---|---|---|---|---|
| MoCo-v2 (Chen et al., 2020b)* | 97.2% | 85.6% | 91.2% | 75.6% | 90.3% |
| SimCLR (Chen et al., 2020a) | 97.7% | 85.9% | 91.3% | 75.9% | 89.2% |
| BYOL (Grill et al., 2020) | 97.8% | 86.1% | 91.6% | 76.3% | 91.7% |
| DINO (Caron et al., 2021) | 97.7% | 86.6% | 91.1% | - | 91.5% |
| SCFS (Song et al., 2023) | 97.8% | 86.7% | 91.6% | - | 91.9% |
| LeOCLR (*ours*) | **98.1%** | **86.9%** | **91.6%** | **76.8%** | **92.1%** |

**Object Detection Task:** To further evaluate the transferability of the learned representation, we compare our approach with other SOTA approaches using object detection on the PASCAL VOC. We follow the same settings as MoCo-v2 (Chen et al., 2020b), fine-tuning on the VOC07+12 trainval dataset using Faster R-CNN with an R50-C4 backbone, and evaluating on VOC07 test dataset. The model is fine-tuned for 24k iterations ($\approx$ 23 epochs). As shown in Table 4, our method outperforms all compared approaches. This superior performance can be attributed to our model's ability to capture richer semantic features compared to the baseline (MoCo-v2) and other approaches, leading to better results in object detection and related tasks.

Table 4: Results (Average Precision) for PASCAL VOC object detection using Faster R-CNN with ResNet-50-C4.

| Approach | $AP_{50}$ | $AP$ | $AP_{75}$ |
|---|---|---|---|
| MoCo-v2 (Chen et al., 2020b) | 82.5% | 57.4% | 64% |
| CLSA (Wang & Qi, 2022) | 83.2% | - | - |
| SCFS (Song et al., 2023) | 83% | 57.4% | 63.6% |
| LeOCLR (*ours*) | **83.2%** | **57.5%** | **64.2%** |

## 5 Ablation Studies

In the following subsections, we further analyze our approach using another contrastive instance discrimination approach (i.e., end-to-end mechanism) to explore how our method performs within this framework. Additionally, we perform studies on the benchmark datasets STL-10 and CIFAR-10 using a different backbone (ResNet-18) to assess the consistency of our approach across various datasets and backbones. Furthermore, we employ a random crop test to simulate natural transformations, such as variations in scale or occlusion of objects in the image, to analyze the robustness of the features learned by our approach, LeOCLR. We also compare our approach with vanilla MoCo-v2 by manipulating their data augmentation techniques to determine which model's performance is more significantly affected by the removal of certain augmentations. In addition, we experiment with different fine-tuning settings to evaluate which model learns better and faster. Furthermore, we adapt the attraction strategy and cropping method of the original image, as well as compute the running time of our approach. Finally, we examine our approach on a non-centric object dataset where the probability of mapping two views containing distinct information is higher.

### 5.1 Different Contrastive Instance Discrimination Framework

We utilize an end-to-end framework in which the two encoders $f_q$ and $f_k$ are updated via backpropagation to train a model with our approach for 200 epochs and 256 batch size. Following this, we conduct a linear evaluation of our model against SimCLR, which also employs an end-to-end mechanism. As shown in Table 5, our approach outperforms vanilla SimCLR by a significant margin of 3.5%, demonstrating its suitability for integration with various contrastive learning frameworks.

Table 5: Comparing vanilla SimCLR with LeOCLR after training our approach 200 epochs on ImageNet-1K.

| Approach | ImageNet-1K |
|---|---|
| SimCLR (Chen et al., 2020a) | 62% |
| LeOCLR (*ours*) | **65.5%** |

### 5.2 Scalability

In Table 6, we evaluate our approach on different datasets (STL-10 and CIFAR-10) using a ResNet-18 backbone to ensure its consistency across various backbones and datasets (i.e., scalability). We pre-trained all the approaches for 800 epochs with batch size 256 on both datasets and then conducted a linear evaluation. Our approach demonstrates superior performance on both datasets compared to all approaches. For instance,

Table 6: SOTA approaches versus LeOCLR on CIFAR-10 and STL-10 with ResNet-18.

| Approach | STL-10 | CIFAR-10 |
|---|---|---|
| MoCo-v2 (Chen et al., 2020b) | 80.08% | 73.88% |
| DINO (Caron et al., 2021) | 84.30% | 78.50% |
| CLSA (Wang & Qi, 2022) | 82.62% | 77.20% |
| BYOL (Grill et al., 2020) | 79.90% | 73.00% |
| LeOCLR (*ours*) | **85.20%** | **79.59%** |

our approach outperforms vanilla MoCo-v2, achieving accuracies of 5.12% and 5.71% on STL-10 and CIFAR-10, respectively.

### 5.3 Center and Random Crop Test

In Table 7, we reported the top-1 accuracy for vanilla MoCo-v2 and our approach after 200 epochs on ImageNet-1K, focusing on two tasks: a) center crop test, similar to (Chen et al., 2020a;b) where images are resized to 256 pixels along the shorter side using bicubic resampling, followed by a $224 \times 224$ center crop; and b) random crop, where images are resized to $256 \times 256$ and then randomly cropped and resized to $224 \times 224$. We obtained the MoCo-v2 center crop result directly from (Chen et al., 2020b), but the random crop result was not reported. To ensure a fair comparison when reporting the random crop results, we replicated MoCo-v2 using the same hyperparameters from the original paper that were used to report the center crop. According to the results, the performance of MoCo-v2 dropped by 4.3% with random cropping, whereas our approach experienced a smaller drop of 2.8%. This suggests that our approach learns better semantic features, demonstrating greater invariance to natural transformations such as occlusion and variations in object scales. Additionally, we compare the performance of CLSA (Wang & Qi, 2022) with our approach, given that both perform similarly after 800 epochs (see Table 1. Note that the CLSA approach uses multi-crop (i.e., five strong and two weak augmentations), while our approach employs only two random crops and the original image. As shown in Table 7, LeOCLR outperforms the CLSA approach by 2.3% after 200 epochs on ImageNet-1K. To address concerns about the increased computational cost associated with training LeOCLR compared to MoCo V2, we include the training time for both approaches in Table 7. We trained both models on three A100 GPUs with 80GB for 200 epochs. Our approach took an additional 13 hours to train over the same number of epochs, but it delivers significantly better performance than the baseline.

Table 7: Comparing LeOCLR with vanilla MoCo-v2 and CLSA after training 200 epochs on ImageNet-1K.

| Approach | Center Crop | Random Crop | Time |
|---|---|---|---|
| MoCo-v2 (Chen et al., 2020b) | 67.5% | 63.2% | 68h |
| CLSA (Wang & Qi, 2022) | 69.4% | - | - |
| LeOCLR (*ours*) | **71.7%** | **68.9%** | 81h |

### 5.4 Augmentation and Fine-tuning

Contrastive instance discrimination approaches are sensitive to the choice of image augmentations (Grill et al., 2020). This sensitivity necessitates further analysis comparing our approach to Moco-v2 (Chen et al., 2020b). These experiments aim to explore which model learns better semantic features and produces more robust representations under different data augmentations. As shown in Figure 4, both models are affected by the removal of certain data augmentations. However, our approach shows a more invariant representation and exhibits less performance degradation due to transformation manipulation compared to vanilla MoCo-v2. For instance, when we apply only random cropping augmentation, the performance of vanilla MoCo-v2 drops by 28 percentage points (from a baseline of 67.5% to 39.5% with only random cropping). In contrast, our approach experiences a decrease of only 25 percentage points (from a baseline of 71.7% to 46.6% with only random cropping). This indicates that our approach learns better semantic features and produces more effective representations for the given objects than vanilla MoCo-v2.

In Table 2, presented in Section 4, we fine-tune the representations over the 1% and 10% ImageNet-1K splits from (Chen et al., 2020a) using the ResNet-50 architecture. In the ablation study, we compare the fine-tuned representations of our approach with the reproduced vanilla MoCo-v2(Chen et al., 2020b) across 1%, 2%, 5%, 10%, 20%, 50%, and 100% of the ImageNet-1K dataset, following the methodology in (Henaff, 2020; Grill et al., 2020). In this setting, we observe that tuning a LeOCLR representation consistently outperforms vanilla MoCo-v2. For instance, Figure 5 (a) demonstrates that LeOCLR fine-tuned with 10% of ImageNet-1K labeled data outperforms vanilla Moco-v2 (Chen et al., 2020b) fine-tuned with 20% of labeled data. This indicates that our approach is advantageous when the labeled data for downstream tasks is limited.

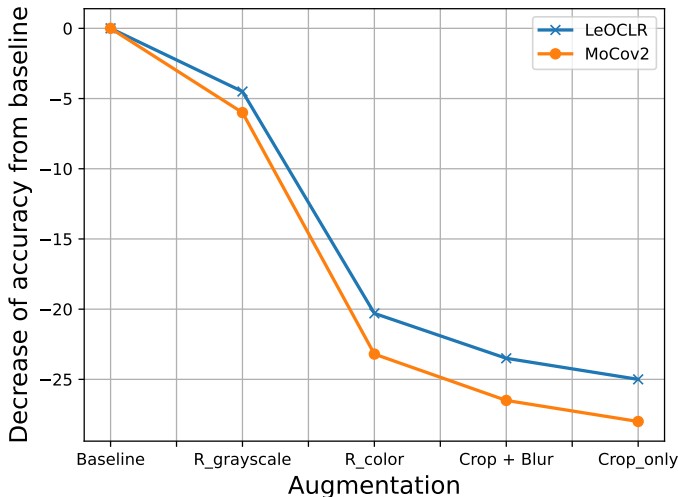

Figure 4: Decrease in top-1 accuracy (in % points) of LeOCLR and our reproduction of vanilla MoCo-v2 after 200 epochs, under linear evaluation on ImageNet-1K. *R_Grayscale* refers to results without grayscale augmentations, while *R_color* refers to results without color jitter but with grayscale augmentations.

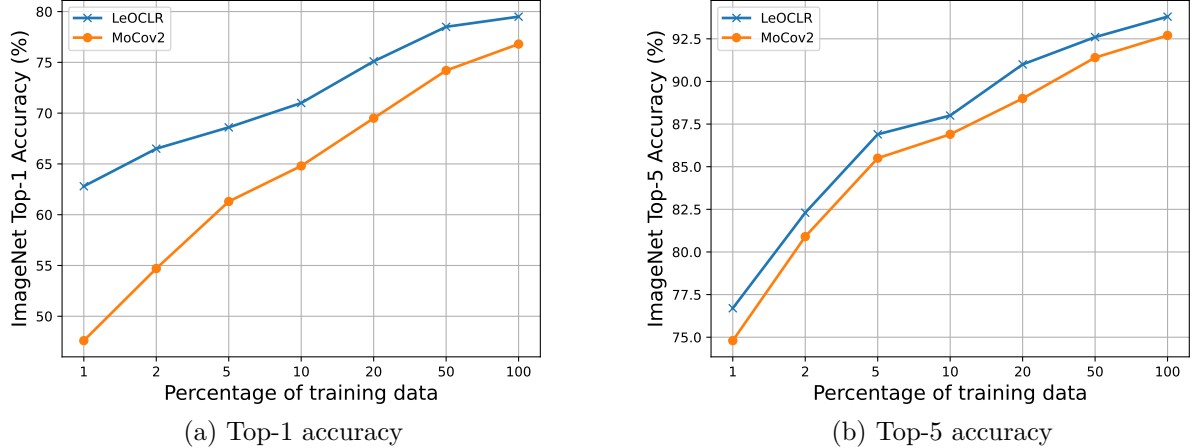

(a) Top-1 accuracy             (b) Top-5 accuracy

Figure 5: Semi-supervised training with a fraction of ImageNet-1K labels on a ResNet-50.

## 5.5 Attraction Strategy

In this subsection, we apply a random crop to the original image $(x)$ and attract the two views $(x^1, x^2)$ toward it to evaluate its impact on our approach's performance. We also conducted an experiment where all views were attracted to each other. However, in our method, we avoid attracting the two views to each other, enforcing the model to draw the two views toward the original image only (i.e., the uncropped image containing semantic features for all crops). For these experiments, we pre-trained the model on ImageNet-1K for 200 epochs using the same hyperparameters employed in the main experiment.

The experiments in Table 8 underscore the significance of the information shared between the two views. They also highlight the importance of leveraging the original image and avoiding the attraction of views containing varied semantic information to preserve the semantic features of the objects. When we create a random crop for the original image $(x)$ and enforce the model to make the two views similar to the

Table 8: Comparisons of augmentation strategies using our proposed approach after 200 epochs.

| Approach | Accuracy |
|---|---|
| LeOCLR (*Random original image*) | 69.3% |
| LeOCLR (*attract all crops*) | 67.7% |
| LeOCLR (*ours*) | **71.7**% |

original image (i.e., LeOCLR(*Random original image*)), the model performance decreases by 2.4%. This drop occurs because cropping the original image and enforcing the model to attract the two views toward it increases the likelihood of having two views with varied semantic information, leading to a loss of semantic features of the objects. The situation worsens when we attract all views $(x, x^1, x^2)$ to each other in LeOCLR (*attract all crops*), causing performance to drop closer to that of vanilla MoCo-v2 (67.5%). This is due to the high probability of attracting two views containing distinct semantic information.

## 5.6 Non-Object-Centric Tasks

Non-object-centric datasets, such as COCO (Lin et al., 2014), portray real scenes where the objects of interest are not centered or prominently situated, as opposed to object-centric datasets like ImageNet-1K. In this case, the probability of creating two views containing distinct semantic information for the object is higher, exacerbating the problem of losing semantic features. Therefore, we train both our approach and the MoCo-v2 baseline from scratch on the COCO dataset to evaluate how our method handles the discarding of semantic features in such datasets. We used the same hyperparameters as for ImageNet-1K, training the models with a batch size of 256 over 500 epochs. Subsequently, we fine-tuned these pre-trained models on the COCO dataset for object detection.

Table 9: Results for pre-training followed by fine-tuning on COCO for object detection using Faster R-CNN with ResNet-50-C4.

| Approach | $AP_{50}$ | $AP$ | $AP_{75}$ |
|---|---|---|---|
| MoCo-v2 (Chen et al., 2020b) | 57.2% | 37.6% | 41.5% |
| LeOCLR (*ours*) | **59.3**% | **39.1**% | **43.0**% |

Table 9 shows that our approach captured better semantic features for the given object than the baseline. This highlights that our method of avoiding the attraction of two distinct views is more effective at preserving semantic features, even in a non-object-centric dataset.

## 6 Conclusion

This paper introduces a novel contrastive instance discrimination approach for SSL to enhance representation learning. Our method mitigates the loss of semantic features by incorporating the original image during training, even when the two views contain distinct semantic content. We demonstrate that our approach consistently improves the representation learning of contrastive instance discrimination in various benchmark datasets, backbones, and mechanisms, such as momentum contrast and end-to-end methods. In linear evaluation, we achieved an accuracy of 76.2% on ImageNet-1K after 800 epochs, outperforming several SOTA instance discrimination SSL approaches. Additionally, we demonstrated the invariance and robustness of our approach across different downstream tasks, including transfer learning and semi-supervised fine-tuning.

## Acknowledgments

We would like to thank the University of Aberdeen's High Performance Computing facility for enabling this work and the anonymous reviewers for their constructive feedback.

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
