# OpenReview forum: "LeOCLR: Leveraging Original Images for Contrastive Learning of Visual Representations"
_TMLR — Accepted by TMLR_

### Review · Reviewer_mf3A · 2024-07-31

**Summary Of Contributions:**

Conventional contrastive learning methods use two randomly cropped and resized images as a positive pair during training. This work argues that these images may contain different semantic information, leading to sub-optimal representation learning. To address this issue, the paper proposes using one original, uncropped image along with a randomly cropped version. Experiments with linear probing and in semi-supervised settings demonstrate that the proposed method outperforms existing approaches on ImageNet and other downstream tasks.

**Audience:**

Yes

**Claims And Evidence:**

No

**Requested Changes:**

There is a ratio parameter when using RandomResizedCrop, which sets the 'lower and upper bounds for the random aspect ratio of the crop.' Can adjusting this parameter address the issue discussed in this paper?

Conventional contrastive methods generate one positive pair per training sample, whereas LeOCLR uses two positive pairs per training sample. The experiments and ablation studies do not rule out the possibility that the improvement from LeOCLR is simply due to using an additional positive pair. Would it be a fairer comparison if other methods also used two positive pairs per training sample? I hope the authors can either justify not performing such a comparison or include additional results on modified baseline methods.

The paper repeatedly uses the terms 'correct' and 'incorrect' semantic positive pairs. There is nothing inherently incorrect about one image focusing on the leg of the dog and another on the head, as both are representations of the dog. The paper mentions occlusion invariance. My understanding is that different views of the dog are precisely how occlusion invariance is achieved. In Figure 1, conventional methods treat all four as positive pairs, and my understanding is that the real issue is that images with different semantic information are treated indiscriminately (also mentioned in Song et al.). I hope the authors can clarify this.

First sentence of Section 3 requires either a reference or should be verified in the paper.

Please carefully review all references, as some of them are either missing information or straight up incorrect: Leverage Your Local and Global Representations: A New Self-Supervised Learning Strategy should be Zhang et al, instead of Qiu et al.

Is $L_t$ in Figure 3 the TotalLoss in (3)? Please ensure that the notation for such crucial definitions is clear throughout the paper.

Please fix the hyperlink at the end of page 5 and at the begining of page 6.

In the second bullet point before Section 4, it is mentioned that there might be time and memory issues with multiple crops. Can the authors verify this claim?

**Strengths And Weaknesses:**

**Strength:**

The proposed method is simple and straightforward to implement.

The review of SSL methods is detailed and well-written.

**Weakness:**

Some explanations for the motivation of the method are unclear (see requested changes).

There is insufficient evidence demonstrating the effectiveness of the proposed method.

The paper appears to lack thorough proofreading, as spelling and grammar mistakes, misuse of capitalization, and incorrect usage of \citep/\citet, and parentheses are frequently observed. The paper could benefit from improved writing and presentation.

---

> ### Author Response · Authors · 2024-09-24
> **Review response to mf3A part (1/3)**
>
> We would like to take this opportunity to thank you for your review of our paper and hope that we can alleviate your concerns below.
>
> There is a ratio parameter when using RandomResizedCrop, which sets the ’lower and upper bounds for the random aspect ratio of the crop.’ Can adjusting this parameter address the issue discussed in this paper?
>
> **In our approach, we used the standard RandomResizedCrop, which is used by the baselines and prior approaches. Predicting the correct random crop ratio for all the Images in ImageNet is not trivial and requires many experiments (trial and error). In addition, even if we found the correct cropping ratio for ImageNet’s images, that wouldn’t guarantee that the same cropping ratio would work with all the datasets. Therefore, the core premise of our paper wouldn’t be solved (i.e., attracting two views containing distinct semantic information). This makes this solution impractical.**
> ***
> Conventional contrastive methods generate one positive pair per training sample, whereas LeOCLR uses two positive pairs per training sample. The experiments and ablation studies do not rule out the possibility that the improvement from LeOCLR is simply due to using an additional positive pair.  Would it be a fairer comparison if other methods also used two positive pairs per training sample? I hope the authors can either justify not performing such a comparison or include additional results on modified baseline methods.
>
> **In our work, we introduce a new attraction strategy based on momentum contrastive to alleviate the loss of semantic features. In our approach, we create two views ($x^1$ and $x^2$) for the same instance (x), similar to the baseline (i.e., MoCo-v2), but we use the original image (x) as a supervisory signal for all the random crops because it contains the semantic information for all the views. The key difference between our approach and the baseline (MoCo-v2) is that the two views are attracted to the original image rather than to each other like the baseline, which led to our proposing a new loss function to accommodate this.**
> ***
> To answer the question, is the performance of our approach achieved because we use two positive pairs, and what will happen if MoCo-v2 uses two positive pairs?
>
> **We would like to illustrate two things:**
>
> **1. If the baseline (MoCo-v2) used two positive pairs for each instance, they would need to create four random cropped views, whereas our approach only uses two views. This means twice the amount of training samples and longer training time.**
>
> **2. In case the baseline used two positive pairs without changing the attraction strategy, the four generated views would still be attracted to each other in the embedding space, which means the main issue of avoiding attracting views that contain distinct semantic information would not be solved. It should be noted that our approach does not attract the two views to each other but, instead, it attracts them to the original image (i.e., image without any crop).**
>
> **Therefore, our approach is not just MoCo-v2 with two positive pairs. It is a new contrastive learning approach based on momentum contrastive and a with new loss function.**
>
> **Finally, we would like to demonstrate that we have made a fair comparison similar to the other multi-crop approaches [4, 5, 3]. We compared our approach with baseline MoCo-v2 because we use momentum contrastive learning to train the model in our approach. Thus, we can prove that our solution enhances the representation learning of contrastive learning and reduces the discarding of the semantic features due to mapping two views that contain two distinct semantic information. After that, we compare our approach with other SOTA approaches as well, including recent multi-crop approaches (i.e., use more than one positive pair for each instance), for a fairer overall comparison (CLSA [5], DINO [4], SCFS [3]).**
> ***

---

> ### Author Response · Authors · 2024-09-24
> **Review response to mf3A part (2/3)**
>
> The paper repeatedly uses the terms ’correct’ and ’incorrect’ semantic positive pairs. There is nothing inherently incorrect about one image focusing on the leg of the dog and another on the head, as both are representations of the dog.
>
> **Some studies have shown that, when using contrastive learning frameworks, two views containing distinct semantic information will inevitably be obtained due to random cropping. This results in the loss of important features for the target objects and thereby degenerates visual representation learning [1, 2, 3].**
> **Since we do not have labels in self-supervised learning, we rely on the information in the shared region between the two views to learn useful representations [1, 2, 6]. In our study, we refer to the views as the ”correct semantic positive pair” when the overlapping or shared region between the two views contains a similar part of the object and the ”incorrect semantic positive pair” when the two views contain distinct parts of the object or a tiny part (this part cannot be used to distinguish this object from other objects) of the object with a large background. We called them incorrect semantic pairs because matching these pairs during training causes discarding semantic features as well as restraining the diversity of the learned representation [3, 6, 2].**
>
> **For example, when we give the model two different human face images and encourage the model to make them have similar representations, the model will focus on the information in the shared region between the two views and extract the general human facial features (i.e., eyes, nose, ears..etc) from the two views and give a similar representation based on this information even if these two images do not belong to the same person. In contrast, let’s assume we have a face image as a first view and a hand image as a second view belonging to the same person, and we encourage the model to make the hand similar to the face; the representation will be something between hand and face, such as skin texture and background information, based on the information in the shared region between the two views. This is what we illustrated in Figure 1: the random cropping might contain incorrect semantic content between the two views even if they belong to the same object. In the revision version we have added ablation studies (subsection 5.1) that demonstrate the importance of avoiding attracting views that do not have similar semantic information.**

---

> ### Author Response · Authors · 2024-09-24
> **Review response to mf3A part (3/3)**
>
> The paper mentions occlusion invariance. My understanding is that different views of the dog are precisely how occlusion invariance is achieved. In Figure 1, conventional methods treat all four as positive pairs, and my understanding is that the real issue is that images with different semantic information are treated indiscriminately (also mentioned in Song et al.). I hope the authors can clarify this.
>
> **Occlusion invariance is achieved if we train the model on variant aspects and scales for the objects in the dataset[1, 2]. To do so, we use random cropping and resizing as data augmentation, but we need to consider the information in overlapping regions or shared regions [1, 2, 6]. In the baseline, if the same object’s part is present in both views (i.e., there is an overlapping region), the model can learn the representation of the object from different scales. Moreover, if we use multi-crops for the same object, the model will learn the representation of the object from different aspects, which yields occlusion invariance. On the contrary, if the two views contain a small part of the dog’s body or different semantic information (i.e., head and leg), which is known as disparate overlapping, the model will discard the semantic features of the object and create a representation based on the small parts of the object that appear in both views [2, 3]. This could lead the model to be sensitive to occlusions and perform poorly when presented with a new image of a dog with a different pose or background.**
> **We argue that our approach is more robust than the baseline in occlusion invariance because we leverage the original image (x) during model training, which contains all the semantic features for the random cropped views. In addition, we refrain from attracting the two views ($x^1$, $x^2$) to each other to prevent the semantic features from discarding while pulling the two views containing the incorrect semantic information. With our approach, all the random crops are attracted to the original image, hence ensuring that shared regions between the original image and the two random crops will always contain correct semantic information. This will preserve the semantic features and allow the model to learn different aspects and scales of the object. In addition, this leads to more occlusion invariance than the baseline. We proved this empirically in Table 7 by comparing our approach with the baseline on random cropped images (i.e., see only part of the object). Our model outperforms the baseline in linear classification by 5.7%. We hope this clarification is helpful.**
>
>
> **References:**
>
> [1] Shlok Mishra, Anshul Shah, Ankan Bansal, Abhyuday Jagannatha, Janit Anjaria, Abhishek Sharma, David Jacobs, and Dilip Krishnan. Object-aware cropping for self-supervised learning. arXiv preprint arXiv:2112.00319, 2021.
>
> [2] Senthil Purushwalkam and Abhinav Gupta. Demystifying contrastive self-supervised learning: Invariances, augmentations and dataset biases. Advances in Neural Information Processing Systems, 33:3407–3418, 2020.
>
> [3] Kaiyou Song, Shan Zhang, Zimeng Luo, Tong Wang, and Jin Xie. Semantics consistent feature search for self-supervised visual representation learning. In Proceedings of the IEEE/CVF International Conference on Computer Vision, pages 16099–16108, 2023.
>
> [4] Mathilde Caron, Hugo Touvron, Ishan Misra, Herv´e J´egou, Julien Mairal, Piotr Bojanowski, and Armand Joulin. Emerging properties in self-supervised vision transformers. In Proceedings of the IEEE/CVF international conference on computer vision, pages 9650–9660, 2021.
>
> [5] Xiao Wang and Guo-Jun Qi. Contrastive learning with stronger augmentations. IEEE transactions on pattern analysis and machine intelligence,45(5):5549–5560, 2022.
>
> [6] Yufei Xu, Qiming Zhang, Jing Zhang, and Dacheng Tao. Regioncl: Exploring contrastive region pairs for self-supervised representation learning. In European conference on computer vision, pages 477–494. Springer, 2022.

---

> > ### Comment · Reviewer_mf3A · 2024-09-24
> >
> > Thank you for the detailed responses, clarifications, and the additional experiment in Sec. 5.1. Most of my concerns have been addressed.

---

> > > ### Author Response · Authors · 2024-09-25
> > >
> > > Thank you for your quick response and acknowledgment. It is much appreciated.

---

### Review · Reviewer_NHig · 2024-08-04

**Summary Of Contributions:**

This paper introduces a novel self-supervised learning approach that addresses the limitations of current contrastive instance discrimination methods. Traditional methods like MoCo-v2 and SimCLR rely heavily on data augmentation, particularly random cropping, which can result in incorrect semantic pairs and degraded representation learning. LeOCLR mitigates this by including the original image in the training process, ensuring that the shared semantic features between augmented views are preserved. This method consistently outperforms existing state-of-the-art approaches across various datasets and downstream tasks, demonstrating superior representation learning.

**Audience:**

Yes

**Claims And Evidence:**

Yes

**Requested Changes:**

Extended Ablation Studies: Include more detailed ablation studies to isolate and evaluate the impact of each component of the proposed method.

Computational Complexity Analysis: Provide a discussion on the computational complexity and scalability of LeOCLR, particularly in comparison to other state-of-the-art methods.

**Strengths And Weaknesses:**

Strength:
Innovative Approach: The introduction of leveraging original images in the training process to preserve semantic features is a significant improvement over traditional methods.
Consistent Performance: The proposed method shows consistent improvement across different datasets and tasks, including linear evaluation, semi-supervised learning, and transfer learning.
Robust Evaluation: The paper provides a comprehensive evaluation of the method, comparing it with various state-of-the-art approaches and demonstrating its effectiveness in different settings.
Clear Methodology: The methodology and the rationale behind the approach are well-explained, making it easier for readers to understand the improvements made.
Weakness:
Limited Ablation Studies: While the paper includes some ablation studies, more extensive ablations could strengthen the claims about the individual contributions of each component of the proposed method.
Scalability Concerns: The paper does not discuss the computational complexity or scalability of the proposed approach in detail, which could be a concern for large-scale applications.

---

> ### Author Response · Authors · 2024-09-24
> **Review response to NHig**
>
> Computational Complexity Analysis: Provide a discussion on the computational complexity and scalability of LeOCLR, particularly in comparison to other state-of-the-art methods.
>
> **We would like to thank the reviewer for enhancing our work. We have added the training time for our work to Table 7. Regarding scalability, we would like to illustrate that we use different architectures and datasets to show consistency on different scales. In Table 6, we used smaller architectures, ResNet18, with smaller datasets (i.e., CIFAR10 and STL10), whereas we used ResNet50 and ImageNet in the main experiments.**
> ***
> Extended Ablation Studies: Include more detailed ablation studies to isolate and evaluate the impact of each component of the proposed method.
>
> **We would like to illustrate that the revised version contains two experiments in the ablation study subsection (5.1). The two experiments provide more analysis of the attraction strategy and the random crop. Notice: the added paragraph font line is blue.**

---

### Review · Reviewer_rwjE · 2024-09-18

**Summary Of Contributions:**

Summary:

unsupervised contrastive learning usually relies heavily on data augmentations. In this paper, the authors suggest that one common augmentation strategy—random cropping—might negatively impact contrastive learning because the two augmented views after cropping might contain different semantic information. As a result, encouraging them to become closer can lead the model to learn incorrect semantic representations. To address this issue, the authors propose a new training/augmentation paradigm, which utilizes the original image as the anchor in contrastive learning. Instead of pulling the two cropped views towards each other, they are pulled towards the anchor image in the latent space. The experimental results show that the proposed training paradigm outperforms several classic contrastive learning frameworks.

**Audience:**

Yes

**Claims And Evidence:**

Yes

**Requested Changes:**

See weaknesses

**Strengths And Weaknesses:**

Strengths:

1. The writing of the paper is clear, with the intuition behind the design of the method well explained and making sense.
2. The research problem is important, as unmeaningful positive pairs can impact the robustness and reliability of the trained model.

Weaknesses:

1. One of the fundamental assumptions of strong augmentations is that the semantic information is not completely distorted. For example, in the illustration, although the two crops capture different parts of the dog, from a human perspective, it is still easy to recognize that the cropped components belong to a dog (i.e., the semantic information is not entirely distorted). The authors should demonstrate why less related positive pairs are not beneficial to contrastive training, as many studies have shown that this is actually advantageous.
2. The proposed method essentially creates a weak augmentation (the original image) and contrasts it with its strong augmentations twice. I fail to see the benefit of doing this.
3. There are prior works that have investigated the problem of semantic distortions during strong augmentations for contrastive learning [1,2]. It is recommended to discuss the relationship and superiority of the proposed method to prior approaches, and to include comparisons in the experiments.

[1] Xiao, T., Wang, X., Efros, A.A. and Darrell, T., What Should Not Be Contrastive in Contrastive Learning. In International Conference on Learning Representations.

[2] Zhang, J. and Ma, K., 2022. Rethinking the augmentation module in contrastive learning: Learning hierarchical augmentation invariance with expanded views. In Proceedings of the IEEE/CVF Conference on Computer Vision and Pattern Recognition (pp. 16650-16659).

---

> ### Author Response · Authors · 2024-09-24
> **Review response to rwjE part (1/2)**
>
> We would like to take this opportunity to thank you for your review of our paper and hope that we can alleviate your concerns below.
>
> One of the fundamental assumptions of strong augmentations is that the semantic information is not completely distorted. For example, in the illustration, although the two crops capture different parts of the dog, from a human perspective, it is still easy to recognize that the cropped components belong to a dog (i.e., the semantic information is not entirely distorted). The authors should demonstrate why less related positive pairs are not beneficial to contrastive training, as many studies have shown that this is actually advantageous.
>
> **The self-supervised approaches rely on the information in the shared region between the views to learn useful representations for the given object because we do not have label as supervised learning [3, 4, 5]. Thus, when we attract the two views to each other, we want the two views to depict similar information for the object in the shared region to capture it during representation learning. For instance, if the first view depicts the dog’s face and the second view contains the dog’s leg, such as in Figure 1 (a) (i.e., the dog’s face is not present in both views), how can the model determine that it should learn the dog’s facial features from the first view and the dog’s leg features from the second and create a good representation for both? In this case, when the objective function forces the model to make the two views similar, the model will focus on the background and the small parts of the object that are shown in the two views (i.e., information in the shared region) to create a representation for the two views. Thus, we gave two different views (i.e., dog’s head and leg), but we ended up with a large representation of the background and a small part of the dog that appears in both views. This yield degrades the model performance on different downstream tasks. We hope this explanation clarifies the case that might cause the random crop to discard semantic features of the object rather than learn better semantic features for the object on different scales and aspects and makes the model more capable of achieving occlusion invariance. Our study emphasizes the importance of random cropping in representation learning. However, we argue that the attracted views should contain similar semantic information to avoid discarding semantic features of the objects during representation learning as we empirically showed, and these studies asserted [6, 5, 4]. Therefore, we introduce a new attraction paradigm that leverages the random crop augmentation correctly, allowing the model to learn different aspects and scales for the objects rather than discarding the semantic information.**
> ***
> The proposed method essentially creates a weak augmentation (the original image) and contrasts it with its strong augmentations twice. I fail to see the benefit of doing this.
>
> **The benefit of applying all data augmentations to the original image (X) except random cropping is to ensure that the shared region between the views (X,$ X^1$,$X^2$) always contains correct semantic information. This is because the (X) view includes all the semantic features for both random crops. In our approach, if two views ($X^1$, $X^2$) contain the same or different semantic information, this will not affect the representation learning because we are not attracting them to each other but attract them to the (X). This is different from baseline (MoCo-v2), where the content of the two views is important to avoid discarding semantic features of the object while attracting them in the embedding space. In the revision version we have added ablation studies (subsection 5.1) that demonstrate the importance of leveraging the original image and avoid attracting views that do not have similar semantic information.**

---

> > ### Author Response · Authors · 2024-09-24
> > **Review response to rwjE part (2/2)**
> >
> > There are prior works that have investigated the problem of semantic distortions during strong augmentations for contrastive learning [1,2]. It is recommended to discuss the relationship and superiority of the proposed method to prior approaches, and to include comparisons in the experiments.
> >
> > **In general, both studies [1,2] focus on the fact that representation invariance due strong augmentations is not always beneficial. In some cases, it might be non-optimal for some downstream tasks. Our work investigates a different issue: the effects of attracting two views containing different semantic information due to random cropping. In [1], they introduced a different issue with the data augmentations pipeline, i.e., they focused on three data augmentations: color, rotation, and texture. They used different embedding spaces, each of which is invariant to all augmentation except one, to prove that one should not use the same augmentation pipeline for all downstream tasks. Our approach differs from [1] in two different ways. Firstly, in our method, we stop random cropping for the (X) view, whereas in [1], they crop all the views. Secondly, in our method, we use the proposed loss function to avoid attracting the views ($X^1$,$X^2$) to each other. Therefore, We did not compare our work with theirs for two reasons:
> > a) they investigated a different augmentation issue, which involved using random cropping for all the views and changed the other data augmentation in the transformation pipeline (i.e., color, texture, and rotation). In contrast, our approach used the same data augmentations for all the views, but we stopped random cropping for one view (X); and b) they conducted their experiments on a subset of ImageNet (ImageNet-100) while our experiments were done on ImageNet-1k. Therefore, the comparison between our works wouldn’t be fair.
> > Regarding [2], they introduce a solution to the effect of representation invariance due to strong augmentations: using hierarchical augmentation invariances. They apply different invariances to different depths of the encoder. We have added their result to Table 1 in the revised version.**
> >
> > **References:**
> >
> > [1] Xiao, T., Wang, X., Efros, A.A. and Darrell, T., What Should Not Be Contrastive in Contrastive Learning. In International Conference on Learning Representations.
> >
> > [2] Zhang, J. and Ma, K., 2022. Rethinking the augmentation module in contrastive learning: Learning hierarchical augmentation invariance with expanded views. In Proceedings of the IEEE/CVF Conference on Computer Vision and Pattern Recognition (pp. 16650-16659).
> >
> > [3] Shlok Mishra, Anshul Shah, Ankan Bansal, Abhyuday Jagannatha, Janit Anjaria, Abhishek Sharma, David Jacobs, and Dilip Krishnan. Object-aware cropping for self-supervised learning. arXiv preprint arXiv:2112.00319, 2021.
> >
> > [4] Senthil Purushwalkam and Abhinav Gupta. Demystifying contrastive self-supervised learning: Invariances, augmentations and dataset biases. Advances in Neural Information Processing Systems, 33:3407–3418, 2020.
> >
> > [5] Yufei Xu, Qiming Zhang, Jing Zhang, and Dacheng Tao. Regioncl: Exploring contrastive region pairs for self-supervised representation learning. In European conference on computer vision, pages 477–494. Springer, 2022.
> >
> > [6] Kaiyou Song, Shan Zhang, Zimeng Luo, Tong Wang, and Jin Xie. Semantics consistent feature search for self-supervised visual representation learning. In Proceedings of the IEEE/CVF International Conference on Computer Vision, pages 16099–16108, 2023.

---

### Decision · Action_Editor_DdzK · 2024-10-11

**Recommendation:** Accept with minor revision

**Comment:**

The paper presents a novel approach to self-supervised learning called LeOCLR that addresses limitations in current contrastive instance discrimination methods. The quality of the work is demonstrated through:

- Experiments across various datasets and tasks
- Performance improvements over state-of-the-art methods
- Some theoretical justification for the proposed approach

*Clarity:*
While the overall methodology is well-explained, there are some areas that could benefit from improved clarity:

- Some explanations of the motivation could be clearer
- The paper would benefit from more thorough proofreading to address spelling, grammar, and formatting issues

*Originality:*
The work shows originality in its approach to preserving semantic features in contrastive learning:

- Novel use of the original, uncropped image in the training process
- New attraction strategy and loss function to address semantic feature loss

*Significance:*
- Consistent improvement over existing methods across multiple tasks and datasets
- Addressing a fundamental issue in contrastive learning approaches
- Potential for broader impact in self-supervised learning applications

*Pros:*

- Novel approach to preserving semantic features in contrastive learning
- Performance improvements across various tasks and datasets
- Some theoretical foundation

Cons:

- Some clarity issues in explanations and writing
- Limited ablation studies
- Lack of detailed computational complexity analysis

*Reasoning behind decision:*

The paper presents a contribution to the field of self-supervised learning. The proposed LeOCLR method addresses a limitation in current contrastive learning approaches and demonstrates consistent performance improvements. The theoretical justification is sound, and the experimental results are convincing. While there are some minor issues with clarity and presentation, these do not significantly detract from the overall quality and importance of the work.

The reviewers' discussions highlight the strengths of the paper in terms of its innovative approach and consistent performance improvements. While there were some concerns raised about clarity and the need for more ablation studies, the authors have addressed many of these issues in their responses.

*Revision*
However, we request that you address the following minor revisions before final publication:

- Improve the clarity of some explanations, particularly regarding the motivation of the method. E.g., a pending request is:

"The authors have made some incorrect claims regarding the distinction between global and local features. While I agree that conducting contrastive learning on a dog's body and its face may not necessarily help the model learn the local features of each specific part, it does indeed aid in learning global features. This is because there are various relationships between a dog's body and its face, such as spatial and topological connections, which contribute to understanding the overall context.

I appreciate the attempt at introducing this novel method; however, based on the authors' explanations, I feel that certain claims in the paper are not sufficiently validated."

- Conduct proofreading to address spelling, grammar, and formatting issues.
- Include a brief discussion on the computational complexity and scalability of LeOCLR compared to other state-of-the-art methods.
- Consider adding ablation studies to isolate and evaluate the impact of each component of the proposed method.

**Audience:**

*Audience*

Yes, the findings of this paper would be of interest to many in the TMLR audience, particularly those working on self-supervised learning, contrastive learning, and representation learning in computer vision.

**Claims And Evidence:**

*Claims*

Yes, the claims are generally well-supported by extensive experimental results across various datasets and tasks. The authors have provided clear evidence of performance improvements over existing methods.

---

> ### Author Response · Authors · 2024-10-12
>
> Thank you for handling our paper and communicating this great outcome. The reviewers’ comments have been useful and have improved our manuscript. We have started preparing the camera ready version and revising it according to your 4 points mentioned above.
>
> Authors

---

> > ### Author Response · Authors · 2024-10-15
> >
> > We thank the reviewers and the editor for handling our paper. Also, we want to illustrate that we made all the required
> >
> > corrections in the camera ready version as follows:
> >
> > **Improve the clarity of some explanations, particularly regarding the motivation of the method (global vs local features).**
> >
> > This is a valid point and we have now incorporated this into our paper. Amended text can be found on page 2 (middle) and in bullet points 1 and 3 on page 6.
> >
> > **Conduct proofreading to address spelling, grammar, and formatting issues.**
> >
> > We have done an extensive proofread of the entire manuscript and reorganised section 5 to present our ablation studies in a clearer manner.
> >
> > **Include a brief discussion on the computational complexity and scalability of LeOCLR compared to other state-of-the-art methods.**
> >
> > We compared the scalability of our approach with different SOTA approaches in Table 6 on page 9 (we also added DINO, CLSA and BYOL). The computational complexity of LeOCLR compared to the baseline is available on page 10, Table 7.
> >
> > **Consider adding ablation studies to isolate and evaluate the impact of each component of the proposed method.**
> >
> > We have added subsection 5.6 to the previous ablation studies section.
> > In this, we compared our approach with the baseline on a non-centric object dataset (COCO), where the object is not located in the center of the image, and the probability of mapping views depicting different semantic information is higher.